# Transparency and Accountability in Sports: Measuring the Social and Financial Performance of Spanish Professional Football

**Rudemarlyn Urdaneta, Juan C. Guevara-Pérez ***[ID]**, Fernando Llena-Macarulla** [ID] **and José M. Moneva** [ID]

Faculty of Economics and Business, University of Zaragoza, 50005 Zaragoza, Spain;
urdanetaruth19@gmail.com (R.U.); fllena@unizar.es (F.L.-M.); jmmoneva@unizar.es (J.M.M.)
* Correspondence: jguevara@unizar.es; Tel.: +34-620-574-171

**Abstract:** This study reviews the impact of the Spanish Transparency Law (TL) 19/2013 and the Union of European Football Associations' (UEFA) Financial Fair Play (FFP) regulations, on the transparency and accountability of Spanish professional football clubs, and examines the influence of financial performance on the transparency of Spanish football clubs. The study uses a Panel Data methodology (FGLS) to compare the international transparency index for football clubs (INFUT) with the criteria of UEFA's FFP as a measure of the social and financial performance, respectively, on a sample of 28 Spanish professional teams of first and second division during the periods of 2015, 2016, and 2019. The study reveals that the implementation of TL 19/2013 and UEFA's FFP has improved the transparency and accountability of clubs. On the one hand, the TL has facilitated access to financial information, and, on the other, the FFP regulations have contributed to improving the balance and financial viability of the clubs. The study also reveals that financial performance directly influences the transparency of clubs.

**Keywords:** transparency; accountability; financial performance; social performance; professional football

## 1. Introduction

A critical difference between sport and business organizations lies in the way they measure performance [1]. The main aim of business organizations is to make a profit, while sport organizations measure performance based on sporting results [2–4].

Thus, the economic profits of a sport organization do not necessarily satisfy its stakeholders if the expected sporting results are not achieved [5].A classic example is European football clubs, in which maximizing performance on the playing field permeates the club's management logic [6].

Over recent decades, the growth of the European football industry and its billion-euro turnover has attracted the interest of major investors, global media and sponsors [6–8].

Various studies confirm that European football clubs often report losses and debts [7,9–12]. Other studies reveal that even investors in European professional football clubs prioritize sports performance over economic performance [13]. In this regard, Buraimo et al. [14] demonstrate that even the most rigid stakeholders are surprisingly patient and tolerant regarding overdrafts and unpaid bills. For fear of the disapproval of the community, banks, investors, creditors, and tax authorities are almost always reluctant to apply strict policies in football clubs to avoid confrontation with fans [15]. Therefore, with the consent of their stakeholders, many clubs continue their operations, despite being bankrupt [6]. This fact urged UEFA regulators to introduce the Financial Fair Play regulation, the scope of which was to introduce more discipline and rationality of clubs' finances and generally to protect the long-term viability of European football clubs [16]. (p. 496). In this sense, Nicoliello and Zampatti [17] point out that the FFP can be considered as the first regulation of European football that tries to transform the traditional approach of "utility maximization" (UM), in

which victories are prioritized in a "profit maximization" (PM) most common approach in the Anglo-Saxon world. In this regard, Wilson et al. [18] found that PM-oriented clubs have better financial health when compared to UM-oriented clubs, and they were more likely to comply with FFP regulations.

In Spain, the tendency to prioritize victories on the field of play (UM) over the Anglo-Saxon model (PM) has been noted. In the early 21st century, Spanish football has reached its competitive peak, claiming two Euro Cups (2008 and 2012) and the 2010 World Championship. The world's greatest footballers play in the Spanish League, and Spanish clubs often occupy the top positions in European competitions and UEFA's coefficient ranking, as well as ranking among UEFA's highest-earning teams [19]. For instance, Real Madrid featured first place in the Money League after generating a record revenue of over €750 m in 2017/18 [20].

However, the situation is very different when it comes to accountability. Over the first decade of the 21st century, Spanish football has accumulated debts and losses to the point that the sustainability of the current business model has been questioned; some clubs have seriously struggled to pay their players, staff, and suppliers, as well as their taxes [21,22].

With this brief description of what the sporting and financial performance of Spanish football has been, we can show the predominance of the UM model characteristic of continental Europe open leagues, and which has already been observed in other leagues in the region, such as the Italian [17]. This structure contrasts with the North American closed leagues in which a PM model predominates. Another distinctive feature can be observed in the previous history and structures of the leagues. For example, English football clubs adopted the status of Sports Corporations (SADs) before the First World War [23], while, in Spain, it is only with Sports Law of 1990 and Royal Decree 1084 of 15 July 1991, which the Spanish Government compelled all professional football clubs with losses to become SADs. Only four clubs in the top two tiers did not become SAD: Athletic de Bilbao, Barcelona, Atlético Osasuna, and Real Madrid. The remaining clubs became SADs at the end of the 1991/92 season [24]. Another difference is the listing on the stock market. While English clubs have been registered since 1983, in Spain, there are no Professional Football League (Liga de Fútbol Profesional (LFP)) clubs to date that have been listed on the Stock Exchange [25], which also allows for less transparency and accountability. In fact, the richest clubs are not Sports Corporations, but Sports Associations (for example: Real Madrid and Barcelona).

With the implementation of the FFP regulations, the literature has focused on assessing the impact of the new regulation on the economic and financial balance [17], the sport's balance [26], or both [27,28]. In this regard, the FFP has not been designed to address the issue of competitive equilibrium [29], as it only limits an individual club's spending relative to its resources [30]. Therefore, this study addresses the financial aspect of the new regulations in Spanish football, the impact of which has been positive compared to other leagues [31].

However, there is a social dimension that becomes more relevant every day in response to the growing demands of stakeholders, and that has not been addressed in-depth due to the complexity of establishing variables capable of measuring these social aspects.

Given the importance of financial information as a measure of transparency, the implementation of Transparency Law (TL) 19/2013 [32] is presented as an attempt to regulate how sport resources are managed and guarantee public access to institutional and economic information pertaining to first- and second-tier clubs that are recipients of public funds, with the purpose of improving communication and accountability. Additionally, Transparency International Spain (TIE) has prepared an international transparency index for football clubs (INFUT), and, given the relevance attributed to transparency by UEFA's FFP [33] and Spanish football Economic Control Regulation (Reglamento de Control Económico (RCE)) [34], it is appropriate to assess the behavior of clubs towards society through their transparency. In this context, the objective of this study is to contrast the influence of financial performance on the transparency of Spanish football clubs.

Thus, the article is organized as follows: after the introduction, Section 2 provides an overview of the theoretical framework and of the link between transparency and financial sustainability in Spanish professional football within the current regulatory framework, before presenting our research hypotheses. The methodology used and the empirical analysis are described in Section 3, and the results are shown in Section 4. Section 5 presents the main findings and discusses some of their implications. Section 6 presents the conclusions.

## 2. Previous Literature

### 2.1. Transparency and Social Performance in Professional Spanish Football

Transparency has been defined as 'clarity in procedures and decision-making, particularly in resources allocation' [35] (p. 31). In the field of sports governance, transparency is becoming increasingly important, and it is the most frequently cited principle in good governance codes for national and international sports organizations [36], as well as being regarded as one of the main tools for evaluating the governance of sports organizations [37].

Two other concepts are often mentioned in the literature in association with transparency: disclosure and accountability. It should be noted that transparency is chiefly understood as a communication process that involves both the availability of information and active participation in the acquisition, creation, and distribution of knowledge [38]. Disclosure is understood as the act of making information about an organization available, either voluntarily or as a result of a legal requirement. In some cases, disclosure and transparency are used interchangeably or in combination [39], but 'just giving information does not constitute transparency' [40] (p. 74). While a certain level of disclosure is necessary to guarantee transparency, disclosure alone is not sufficient to ensure high standards of transparency, for disclosure can decrease transparency if an excess of information is used to disguise important facts [41].

On the other hand, Spanish clubs going into administration, tax defaults, and lack of transparency have projected a bad image of the sports sector, a loss of reputation, and understandable public suspicion [42]. In this regard, the implementation of TL 19/2013, is presented as a regulatory action on the part of the state to alleviate public distrust by facilitating public access to institutional and economic information [32] pertaining to first- and second-tier clubs that receive public financial support, and by strengthening communication channels and accountability.

In Spain, Transparency International (TI) has developed a specific transparency index for football clubs (INFUT). This index comprises 60 indicators divided into five (5) areas of transparency that include information about the club (INCLUB), relations with members, fans and the general public (INSOC), economic-financial transparency (INEF), transparency in contracts and supplies (INCS), and, finally, the indicators set out in TL 19/2013 (INT) [43].

The report of the third edition of INFUT in 2019 reflects that the number of clubs that manage to be 100% transparent increased from 4.88% in the first two editions to 45.24% in the third edition. In the same way, the clubs that achieve an efficiency greater than 80% increased from 7.32% in 2015 to 26.83% in 2016, reaching up to 90.48% of the clubs in 2019. Additionally, for 2015, 58.54% of the clubs were below 50% transparency. This proportion was reduced to 26.83% in the second edition, and dropping to 0% in the third edition [43].

INFUT suggests that, on average, club transparency increased from 44.21% in 2015 to 62.81% in 2016 and, finally, to 93.63% in 2019. For greater detail, Table 1 presents the descriptive statistics of the 5 transparency areas outlined in TIE's reports by year [43]. As shown, the economic-financial and contracts and supplies areas yield the lowest results. This is consistent with Pielke's [8] results, which suggest that, among Grant and Keohane's [44] seven mechanisms of accountability, the interaction between fiscal accountability and legal accountability is the most important in the football industry. Therefore, it seems reasonable that both UEFA's FFP and the Spanish LPF's RCE focus on these aspects.

**Table 1.** Descriptive statistics for transparency areas.

| | Min | | | Max | | | Mean | | | Std. Dev. | | |
|---|---|---|---|---|---|---|---|---|---|---|---|---|
| | 2015 | 2016 | 2019 | 2015 | 2016 | 2019 | 2015 | 2016 | 2019 | 2015 | 2016 | 2019 |
| INFUT | 10.00 | 15.80 | 53.13 | 100.00 | 100.00 | 100.00 | 44.21 | 62.81 | 93.63 | 23.23 | 23.39 | 10.68 |
| INCLUB | 22.20 | 16.70 | 65.00 | 100.00 | 100.00 | 100.00 | 51.21 | 75.73 | 94.17 | 23.04 | 23.29 | 9.17 |
| INSOC | 16.70 | 30.60 | 64.29 | 100.00 | 100.00 | 100.00 | 51.34 | 74.11 | 96.68 | 22.91 | 21.89 | 6.94 |
| INEF | 0.00 | 0.00 | 0.00 | 100.00 | 100.00 | 100.00 | 34.24 | 49.64 | 90.60 | 30.60 | 33.84 | 22.64 |
| INCS | 0.00 | 0.00 | 0.00 | 100.00 | 100.00 | 100.00 | 22.44 | 40.73 | 87.62 | 28.00 | 33.94 | 24.18 |
| INT | 0.00 | 0.00 | 0.00 | 100.00 | 100.00 | 100.00 | 52.10 | 64.30 | 94.18 | 27.86 | 24.65 | 16.87 |

INFUT: General transparency index for football clubs, INCLUB: information about the club sub-index, INSOC: relations with members, fans and the general public sub-index, INEF: economic-financial transparency sub-index, INCS: transparency in contracts and supplies sub-index, INT: indicators set out in TL 19/2013 sub-index.

The difficulty of measuring the vigor of the social management of sports organizations in continental Europe concerning Anglo-Saxon environments is a consequence of the marked differences between both sports systems when communicating their social responsibility strategies. While, in some European countries, these actions are expressed implicitly, in Anglo-Saxon countries, it is explicit [45]. In this sense, given the importance that the regulation attributes to transparency, the INFUT is presented as a proxy variable for club's social performance.

*2.2. Accountability and Financial Performance in Professional Football*

When information concerning the actions of organizations is released, there is always overlap in the use of the term accountability and transparency. The literature reveals that accountability is a broader concept, in how organizations are held responsible for their actions [46], while transparency focuses on providing accurate, understandable, and accessible information to stakeholders [37].

In this context, the persistent behavior of professional football teams to consider sporting results above profit [2–4] often leads to financial hardship [7,9–12].

To guarantee the financial viability of clubs, the governing body, UEFA, issued the FFP in 2010. UEFA controls the financial status and performance of clubs based on accounting data. Clubs are expected to present 'break-even results' (equal revenue and expenses [33,47,48]. Clubs that fail to meet these standards face strict penalties, such as fines or disqualification from the European club competitions [49]. Moreover, the inability to meet the financial thresholds set by the FFP also leads to a loss of income, putting the financial viability of most clubs at risk. Deloitte's reports [10] expect that the advent of the FFP is to lead to a change in philosophy in many clubs, prompting them to adopt more balanced management practices [33].

In Spain, Sánchez et al. [50] point out that the Sports Law enacted in 1990 already gave the LFP the power to supervise the finances of professional clubs. In line with FFP regulations, in 2014, the LFP introduced additional financial controls that resulted in more rigorous supervision, both concerning previous seasons (a posteriori) and budgets for future seasons (a priori) Economic Control Regulation (RCE) of Sports Associations ((Asociaciones Deportivas (ADs)) and Sports Public Limited Companies (Sociedades Anónimas Deportivas (SADs)), affiliated with the Spanish LFP. The general aim of these regulations is 'to promote the solvency of ADs and SADs' [34]. In addition, these regulations also aim 'to increase the economic and financial capacity of clubs as well as their transparency' [34].

However, some studies suggest that this regulatory effort may not be having the intended results, reducing, rather than increasing, the transparency and credibility of clubs' managers, since the imposition of regulatory control linked to reporting based on accounting data can lead to a deterioration in accounting quality [15], to a discretionary choice of the audit firm [51], and an increase in auditing fees [25].

However, the financial reports of the Higher Sports Council of Spain (Consejo Superior de Deportes (CSD)) [52] present evidence of the improvements in the financial situation

of Spanish football as a consequence of the new regulations. Mareque et al. [25] observe improvements in the financial performance of the clubs, and Ahtiainen and Jarva [31] highlight the positive impact of the regulations on Spanish football compared to other European leagues. This behavior is related to that observed in other sectors of Spanish sport in response to new financial regulations [53].

On average, revenues and expenses increase, but the latter to a lesser degree, and this has contributed to improving their earnings, which meets the expectations of the UEFA regulations by seeking that clubs present 'break-even results' (equal revenues and expenses) [33,47,48]. Figure 1 shows the behavior of the revenues and expenses of the Spanish first division during the last 20 years.

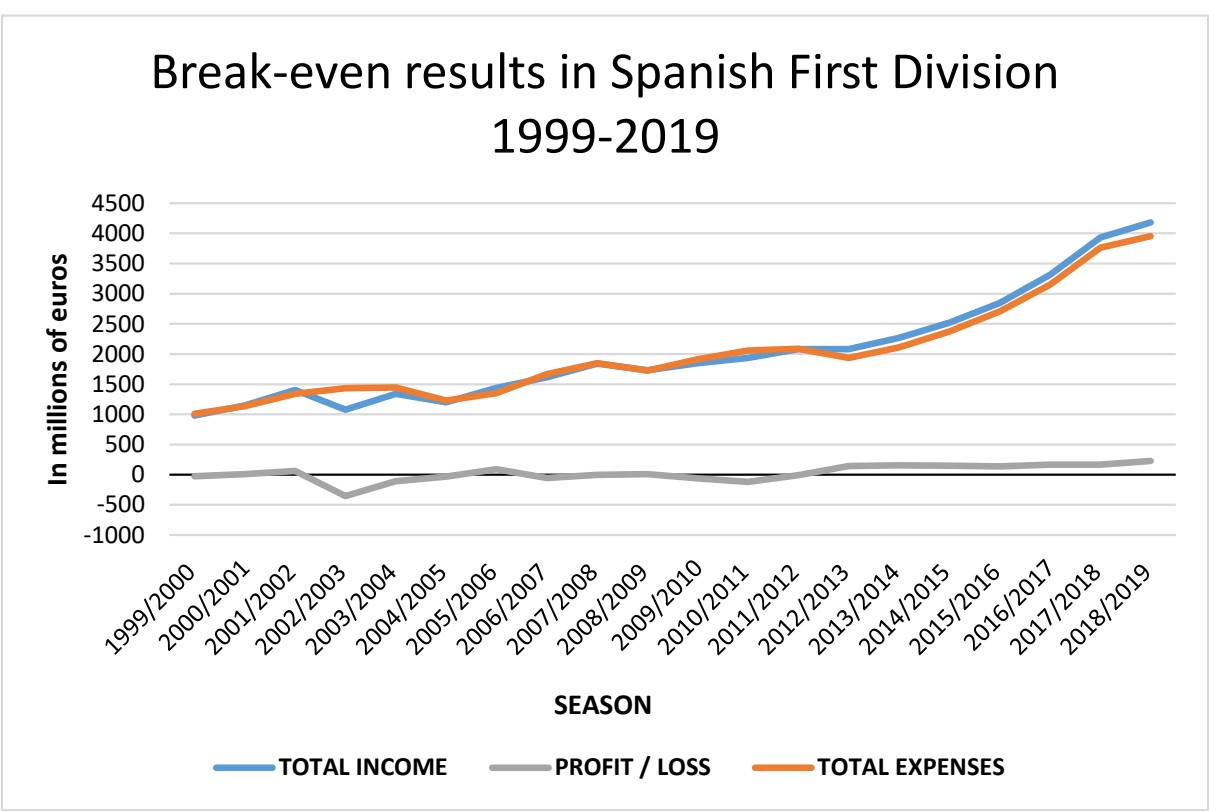

**Figure 1.** Break-even results in Spanish First Division 1999–2019. Source: CSD. Sub directorate General for Professional Sports and Financial Control. Balance of the Economic-Financial Situation of Spanish Football 1999–2019 [52].

As can be seen, revenues and expenses are balanced just at the time the new regulations are implemented, and benefits begin to be observed.

### 2.3. Social Performance vs. Financial Performance in Spanish Professional Football

At present, research on football clubs has only been approached from a financial and sports perspective. However, there is also a social dimension of public relevance for professional football clubs that are gaining importance, which makes the contrast between transparency and accountability of football clubs in response to TL 19/2013 and UEFA's FFP (as measures of social and financial performance, respectively) require a theoretical framework on which to base the analyses.

In this regard, the relationship between social performance and financial performance has been considered in theoretical and empirical studies, in which a theoretical framework [54] arises as a response to the skepticism of the neoclassical argument subscribed to the utility paradigm, according to which companies have only one social responsibility: to increase their profits [55]. In sports, this objective is replaced by obtaining competitive

results. Such is the case of professional football, where these two management models, UM and PM, come into the debate.

The theoretical frameworks that have emerged in subsequent decades are determined by the sign (positive, negative or neutral) and the causal sequence of the relationship, according to which social actions can be contrasted as an independent, dependent variable, or that there is a synergistic link [56]. All these possible theoretical relationships have been worked with a linear model, which is summarized in Table 2.

**Table 2.** Typologies of the relationship between SP and FP.

| CAUSAL SEQUENCE | SIGN OF THE RELATIONSHIP | | |
| --- | --- | --- | --- |
| | Positive | Neutral | Negative |
| SP →FP | Social impact hypothesis | | Trade-off hypothesis |
| FP →SP | Hypothesis of the availability of funds | Hypothesis of the "moderator" variables | Hypothesis of opportunism of managers |
| SP ↔ FP | Positive synergy | | Negative synergy |

Source: Preston and O'Bannon [56].

To contrast the previous relationships, among the empirical studies that try to confirm the sign of the relationship and the causal sequence between these variables, the multivariate analyses that examine the links between different measures of the social and financial variables are highlighted, incorporating the influence of moderating factors, such as the size of the company, sector, I + D + i, etc. [57].

Through the use of the predominant ordinary least squares methodology in the literature, the most common model is reduced to the following expressions:

$$SP = \alpha + \beta FP + \delta CV + \varepsilon, \tag{1}$$

$$FP = \alpha + \beta SP + \delta CV + \varepsilon, \tag{2}$$

where social performance SP will be the dependent variable, the financial performance FP the independent variable (1), or vice versa (2); CV represents the control variables that would moderate the SP-FP relationship, or vice versa; and $\varepsilon$ is the error term.

When reviewing the empirical research [57–59], the trade-off hypothesis seems to be rejected in favor of the social impact hypothesis; in the same way, the hypothesis of the opportunism of managers is displaced by the hypothesis of the availability of funds, indirectly rejecting the hypothesis of negative synergy (see Table 2). There is little contribution to the hypothesis of positive synergy since few studies consider the relationship between Social and Financial Performance in both directions.

Some studies have contrasted the SP-FP binomial on European samples [60] and Spanish samples [61]. However, most of the studies are based on American or English companies, which makes it necessary to research European companies, in general, and Spanish companies, in particular [57–62]. In addition, the usual has been cross-sectional studies located in different industries, having greater consistency if they focused on a specific sector [63]. In this regard, only the study by Inoue, Kent and Lee [64] stands out, who focused on sports, analyzing the main professional leagues in the United States, with the limitation of "mixing" different sports. All this justifies that this study is framed in the Spanish context and a specific sector: professional football.

*2.4. Formulation of Hypotheses*

In the framework of the hypothesis of the availability of funds [56], the present study contrasts the influence of financial performance on the transparency of Spanish football clubs. For this, a set of specific hypotheses could confirm in greater detail the behavior of the SP-FP relationship.

Generally, financial performance is measured with ratios, such as financial profitability (Return on Equity- ROE) and, less often, economic profitability (Return on Asset-

ROA). However, in the football industry, the number of clubs that usually report negative net worth undermines comparisons between clubs, making this ratio somewhat meaningless [65]. For this reason, since the denominator in ROA represents the value of the club's total assets that never assume negative values, the specialized literature has tended to use ROA to measure the performance of Spanish [21,25], and European, football clubs [15,16,51,65,66].

Since regulators seek 'to improve the economic and financial capacity of the clubs, increasing their transparency and credibility' [33,34], our first hypothesis is formulated as follows.

**Hypothesis 1 (H1)**. *ROA is directly associated with the levels of transparency in football clubs.*

According to Dimitropoulos and Tsagkanos [66], European football clubs seem to be more win maximizers than profit maximizers and are willing to resolve on debt financing and sustain severe losses for enhancing their on-field performance. Therefore, we believe that leverage (as measured by the ratio of total debt to common equity) will have a negative relation with transparency and the tendency to report small positive income, formulating our second hypothesis as follows.

**Hypothesis 2 (H2)**. *Leverage is negatively associated with the levels of transparency in football clubs.*

On the other hand, Kelly [67] points out that in the football industry the manager's target is to win championships, avoid relegations, and remain solvent. In this regard, some studies show that clubs prioritize solvency over profitability [68], perhaps, intending to change possible mimetic isomorphisms that have become institutionalized in the sector, such as increasing debts and persistent deficits [7,9–12], which have led popular clubs to the brink of bankruptcy [6].

In addition to this, solvency is a priority criterion for sport governing bodies and the reason why internal regulations of some European Leagues, such as Spanish LFP [34], or the Direction Nationale du Contrôle de Gestion (DNCG) French [69], give priority to solvency over other financial ratios when it comes to allocating resources. For these reasons, our third hypothesis is formulated as follows.

**Hypothesis 3 (H3).** *Solvency is positively associated with the levels of transparency in football clubs.*

In addition, we consider that the behavior of the clubs towards their creditors should be valued both in the long term and in the short term. This fact leads us to the need to confirm whether the liquidity of the clubs is a determinant of transparency and, along the same lines, to establish our fourth hypothesis.

**Hypothesis 4 (H4).** *Liquidity is positively associated with the levels of transparency in football clubs.*

Dependence on a single source of income has an impact on accountability, compromising the financial sustainability of clubs. Such is the case of clubs that are overly reliant on broadcasting rights, a widespread phenomenon in European football [50]. We believe that these clubs can even compromise their transparency to meet the conditions set by UEFA, LFP, and other regulatory bodies and guarantee their main source of income since the dependence on resources usually leads to acting coercively [53]. Because clubs are demonstrably prone to commit beyond their means, the regulations demand that they 'operate according to income' [34]. We believe that adequate segregation of income would help to mitigate these risks. For this reason, our fifth hypothesis is formulated as follows.

**Hypothesis 5 (H5).** *Income concentration is indirectly associated with the level of transparency of football clubs.*

Regarding our control variables, according to Orlitzky [70], the size of companies is positively related to business performance and viability because it can lead to economies of scale and greater control of resources by stakeholders. Additionally, the potential for regulatory scrutiny increases as companies become larger and more profitable [71,72]. There are previous studies on clubs from major European professional football leagues that confirm this behavior [15], and others that reject it [69], so this relationship is not clear in the football industry.

On the other hand, the results of Transparency International highlights that 'transparency is a matter of will or attitude, rather than of economic size or budget capacity'; in 2015, the two most transparent first-tier clubs were at the opposite ends of the budget bracket (Real Madrid: € 578 million; Eibar: € 16 million), and the four most transparent clubs in 2016 were relatively small in terms of economic size [43], behavior that is replicated for 2019. In the framework of these contradictory statements, we have formulated our sixth hypothesis as follows.

**Hypothesis 6 (H6).** *Size is positively associated with the levels of transparency in football clubs.*

Finally, the need arises to consider whether the observed changes attend to a maturing process of football clubs as a consequence of the new regulations. For this reason, we have formulated our seventh and last hypothesis.

**Hypothesis 7 (H7).** *Time is positively associated with the levels of transparency in football clubs.*

As a whole, the proposed hypotheses will allow us to contrast whether financial performance has an impact on the transparency of the clubs and, with it, the traditional hypothesis of the availability of funds.

## 3. Materials and Methods

### 3.1. Sample and Data

The sample initially comprised the 42 clubs in the top two Spanish professional football leagues; clubs that did not provide full financial reports or transparency indexes for the period under study (2015, 2016, and 2019) or that were relegated to the third tier in any of these two seasons were excluded. Finally, we have obtained a balanced panel of 28 clubs ($n = 28$), of which four (4) are Ads, and the remaining (24) SADs, during the 3 seasons ($t = 3$) in which TIE published its INFUT indexes (2014–2015, 2015–2016, 2018–2019), for a total of 84 observations. The financial data has been extracted from the Iberian Balance Analysis System database (SABI) and the transparency portals on the club's websites.

### 3.2. Empirical Analysis

In order to assess whether the performance and financial sustainability of football clubs positively influence transparency, and to test our research hypotheses, we have incorporated transparency as a dependent variable in the following model:

$$\text{INFUT}_{it} = \alpha_0 + \alpha_1\text{ROA}_{it} + \alpha_2\text{LEV}_{it} + \alpha_3\text{SOLV}_{it} + \alpha_4\text{LIQ}_{it} + \alpha_5\text{HI}_{it\text{-}1} + \alpha_6\text{SIZE}_{it} + \alpha_7\text{TIME}_{it} + \varepsilon, \quad (3)$$

where the INFUTit variable represents the Transparency Index of Football Clubs prepared by TIE.

McGuire et al. [73] argue that accounting data, and especially ROA, are generally better indicators of financial performance than measures based on market values, such as the Altman's Z-score or Tobin's Q, which must be ruled out because no Spanish football club is listed on the stock market. Therefore, following previous studies [21,25], we have used ROA (represented by operating income before taxes and interest divided by total assets) to test our first hypothesis (H1).

Given that one of the objectives of the FFP is to reduce debt, the variable leverage (LEVit) measured by the ratio of total debt to common equity contrasts our (H2) based on the percentage of external financing of the teams, since the clubs more leveraged will be further from complying with the regulations, and, consequently, they will seek to be less transparency so as not to be sanctioned.

Since the general objective of the LFP's RCE (a posteriori) is to promote the solvency of ADs and SADs [34], the SOLVit solvency ratio has been used (ratio of total assets over total liabilities), as well as liquidity (ratio of current assets to current liabilities), to test our third (H3) and fourth (H4) hypotheses. These ratios measure long- and short-term business viability based on strict book values and, in turn, allow comparisons between clubs of different sizes. Dimitropoulos and Tsagkanos [66] used both accounting measures, financial performance (ROAit) and business viability (SOLVit), in their analysis of the European football sector.

To test our fifth hypothesis (H5), the HIit variable was considered as an expression of the Herfindahl index (H Index). This indicator is used in different areas as a measure of concentration; therefore, it allows us to assess income concentration, as well as clubs' degree of financial dependence on specific sources of income. The H index is the sum of the squares of the percentage of total income represented by each of the top four main sources of relevant revenues, according to the UEFA's FFP [33]: Broadcasting rights, sponsorship and advertising, gate receipts, and commercial activities, according to the club's financial statements. This is expressed as follows:

$$H = \sum_{i=1}^{m} p_i^2, \tag{4}$$

where $H$ represents the $H$ Index, $i$ the four (4) main sources of income, and $pi$ the percentage of each club's income represented by the source of income $i$. In this occasion, the $H$ Index will oscillate between 0.25 and 1; $H = 1$ indicates maximum income concentration, Hand 0.25 maximum income dispersion.

The H Index represents a proportion of an accounting item of a nominal nature, such as income (unlike the rest of the financial variables described so far, which are based on accounting ratios over real items). In this sense, because the regulations demand that they 'operate according to income' [34], it is logical to expect that the application of said income will be reflected in the following season. In this regard, Birkhäuser, Kaserer, and Urban [74] present evidence that, after the introduction of FFP, former season's winners are correlated with greater budget shares in the upcoming season. Based on these arguments, we consider it appropriate to incorporate a lagged effect in the H Index variable considering the previous year, as in preceding studies [31].

Additionally, the model incorporates two control variables. First, to test our sixth hypothesis (H6), model (1) considers the size (SIZEit) of clubs, a variable that has proven to be an important factor concerning the quality of accounting in previous research in the sports field [15,16]. We shall use the natural logarithm of total assets (LnTA) at the end of the fiscal year as a proxy variable (SIZEit) for club size, a widespread method to normalize potentially biased values in financial studies [53]. In this regard, larger clubs are expected to be more transparent, as the effect of regulatory scrutiny increases as companies become larger [71,72], although INFUT 2015 and 2016 claim the opposite [43]. No sign can be inferred for the coefficients of H4, owing to the contradictory empirical evidence discussed in the previous section.

Second, to test our fifth and last hypothesis (H7), we have considered adding to the model the variable TIMEit, with which it is intended to capture the temporal effect of the learning process and adaptation to the new measures. Table 3 presents the relationship between regulatory frameworks and the respective variables and indicators.

**Table 3.** Operationalization of the variables.

| Regulatory Framework | Performance | Variables | Indicators |
|---|---|---|---|
| TL 19/2013 | Social Performance | Transparency | Index INFUT |
| UEFA's FFP/LFP's RCE | Financial Performance | Accountability | ROA/Leverage/Solvency/ Liquidity/Herfindahl Index |

Source: Author's elaboration.

## 4. Results

Model (3) has been operationalized from a balanced panel of 28 clubs (*n* = 28) during the 3 seasons (*t* = 3) in which TIE has published its INFUT indices (2014–2015, 2015–2016, 2018–2019), for a total of 84 observations. The model was also tested using the Doornik-Hansen normality test [75]. The findings indicate that all the residuals in the model have a normal distribution, with a significance level of 1%.

When observing the descriptive statistics (2015–2016–2019) for Equation (3), we see that, on average, the transparency of the clubs is 70.44%. For the financial variables in Table 4, we observe some stability in the ROA > 5% variable. However, the minimum values suggest that a significant number of clubs has low profitability. This same situation can be observed in leverage (>1) and solvency (>1.5), which, on average, seem stable, but the wide gap between minimum and maximum values shows a significant imbalance in the financial balance of the clubs. On average, liquidity presents a low capacity of the clubs to honor their commitments in the short term (<1.5). On the other hand, the H Index suggests a moderate concentration, the average being 0.475. All of this highlights the efforts made by clubs to find alternative sources of revenue, possibly as a result of the need to comply with the FFP's break-even rules.

**Table 4.** Descriptive statistics (2015–2016–2019) for Equation (3).

| Variable | Min | Max | Mean | Std. Dev. |
|---|---|---|---|---|
| INFUT | 10.000 | 100.000 | 70.436 | 28.594 |
| ROA | −35.276 | 45.625 | 6.884 | 13.945 |
| LEV | −26.344 | 22.703 | 2.816 | 6.697 |
| SOLV | 0.222 | 8.361 | 1.749 | 1.378 |
| LIQ | 0.057 | 6.761 | 0.977 | 0.984 |
| INDICE H t-1 | 0.253 | 1.000 | 0.475 | 0.176 |
| LOG ACT | 15.291 | 21.030 | 18.018 | 1.380 |
| TIME | 1.000 | 3.000 | 2.000 | 0.821 |

Source: Author's elaboration.

In the periods analyzed, the correlation between the variables in Equation (3) suggests that the transparency of the clubs increases over time and that the larger clubs are more leveraged, less solvent, have less liquidity, and of a lower concentration of income. It also seems that solvency and liquidity are related to each other, and both to the income concentration of the clubs. However, the variance-inflation factor test indicates the absence of collinearity (vif < 10), therefore that none of these correlations is particularly strong. Table 5 shows globally the correlations between the analyzed periods:

**Table 5.** Pearson Correlation matrix (2015–2016–2019) variables in Equation (3).

| Variables | INFUT | ROA | LEV | SOLV | LIQ | IH t-1 | SIZE | TIME |
|---|---|---|---|---|---|---|---|---|
| INFUT | **1** | | | | | | | |
| ROA | 0.135 | **1** | | | | | | |
| LEV | −0.068 | −0.057 | **1** | | | | | |
| SOLV | 0.179 | −0.048 | −0.123 | **1** | | | | |
| LIQ | 0.155 | −0.09 | −0.127 | **0.728** | **1** | | | |
| IH t-1 | −0.029 | −0.144 | −0.101 | **0.276** | **0.416** | **1** | | |
| SIZE | 0.153 | −0.035 | **0.31** | **−0.321** | **−0.302** | **−0.295** | **1** | |
| TIME | **0.725** | −0.023 | 0.007 | 0.09 | 0.113 | 0.076 | 0.143 | **1** |

Note: Values in bold are different from 0, with alpha significance level = 0.05.

First, traditional OLS techniques (OLS) with random effects and fixed effects are evaluated. To discriminate between these two models, a Hausman test was performed, which yielded a Chi-square = 3.47, corresponding to a *p*-value = 0.7480, which would be recommending the estimates of random effects, something that is also consistent with the data structure of the sample (N > T). However, as can be seen in Model 1 of Table 6, the results obtained are not entirely favorable. Therefore, it is necessary to verify some possible problems, such as cross-sectional dependence and heteroscedasticity. First, we have ruled out cross-sectional dependence problems using the Pesaran [76] test by obtaining a statistic of 1.087 and a *p*-value = 0.2772. To verify heteroskedasticity, we perform the Breusch-Pagan and Cook-Weisberg test [77] for heteroskedasticity in a linear regression model, the results of which indicate a heteroskedasticity problem (chi2 = 8.91 and Prob > chi2 = 0.0028).

**Table 6.** Results of the estimation of Equation (3).

| | MODEL 1 Random-Effects GLS Regression | | | MODEL 2 FGLS Regression | | |
|---|---|---|---|---|---|---|
| | Coefficient | z | Significance | Coefficient | z | Significance |
| $ROA_{it}$ | 0.3182122 | 2.01 | 0.044 ** | 0.361393 | 7.59 | 0.000 *** |
| $LEV_{it}$ | −0.3610613 | −1.09 | 0.275 | −0.485518 | −3.49 | 0.000 *** |
| $SOLV_{it}$ | 3.513109 | 1.47 | 0.142 | 2.755744 | 3.69 | 0.000 *** |
| $LIQ_{it}$ | 0.0089839 | 0.00 | 0.998 | 2.173572 | 2.03 | 0.043 ** |
| $IH_{it-1}$ | −10.72099 | −0.79 | 0.428 | −16.40291 | −4.42 | 0.000 *** |
| $SIZE_{it}$ | 2.519967 | 1.25 | 0.213 | 2.160404 | 2.77 | 0.006 ** |
| $TIME_{it}$ | 24.43574 | 10.23 | 0.000 *** | 24.03663 | 21.65 | 0.000 *** |
| Constant | −26.06973 | −0.69 | 0.493 | −17.98682 | −1.43 | 0.152 |
| | R-sq = | | 0.5869 | Time periods = | | 3 |
| | Number Observations = | | 84 | Number Observations = | | 84 |
| | Number of groups = | | 28 | Number of groups = | | 28 |

Note: (***), (**) indicate statistically significant coefficients at the 1% and 5% levels, respectively.

Faced with this situation, the Feasible Generalized Least Squares (FGLS) model presents a solution to this problem, which is why it has been the selected model (Model 2) to perform the estimates of Equation (3), and whose results are those that we finally present in Table 6.

When the impact of financial performance on transparency is analyzed, the first thing that stands out is a positive and statistically significant relationship (*p* = 0.000 ***) between ROA and transparency. This suggests that more profitable clubs be more transparent. In this sense, our first hypothesis is accepted (H1). Regarding Leverage, a negative and statistically significant coefficient (*p* = 0.000 ***) is observed, confirming our fourth hypothesis (H2), according to which the most leveraged clubs are less transparent.

Additionally, the results show a positive and statistically significant relationship between transparency and the ability of clubs to honor their commitments in the short term (LIQ *p* = 0.043 **) and long term (SOLV *p* = 0.000 ***), which confirms our second

(H3) and third hypotheses (H4). This result shows that the transparency of football clubs is conditioned by their financial sustainability.

Regarding the relationship with the H Index, the results show a negative and statistically significant relation ($p = 0.000$ ***), confirming our fifth hypothesis (H5), according to which the clubs with the highest concentration of income are expected to be less transparent, and vice versa.

When observing the control variables of TIME and SIZE, the results confirm, in the first place, a positive effect of time on transparency, which, in turn, justifies the heteroscedasticity observed in the 1 OLS model. Finally, concerning size, the sign of the coefficient determines that the largest clubs (SIZEit) would be more transparent. The results are statistically significant for both cases, (TIME $p = 0.000$ *** and SIZE $p = 0.006$ **), which leads us to accept the sixth (H6) and seventh (H7) hypotheses.

The general results confirm that the financial performance of the clubs influences transparency and, with it, the hypothesis of the availability of funds [56].

## 5. Discussion

The study contributes to regulators a look at the impact of TL 19/2013 and UEFA's FFP regulations on the transparency and accountability of Spanish professional football clubs. Second, a set of specific hypotheses has allowed us to test whether financial performance affects the transparency of clubs and, with it, the traditional hypothesis of availability of funds [56].

In this sense, the study contributes a European vision to the traditionally English-speaking countries approach predominant in the specialized literature that contrasts Social and Financial Performance [57,62]. This fact attributes a relevance to the present study, given the marked differences between Anglo-Saxon law and continental European law when assessing the impact of introducing new regulations in sport. On the other hand, by focusing on professional football and analyzing different periods, a different perspective is offered than the usual cross-sectional studies that contrast social and financial performance on samples from different industries [63] or between various sports [64].

An additional contribution of this study has been the incorporation of an ad hoc index for Spanish professional football clubs (INFUT), which manages to group 60 indicators in 5 areas of transparency as a robust measure of social performance that overcomes the limitations observed in previous studies [64]. On the other hand, accountability determines financial performance based on UEFA's financial fair play criteria and the LFP's RCE. As such, following Dimitropoulos and Tsagkanos [66], we have used economic profitability (ROA) and solvency values to measure the clubs 'financial performance and business viability. Leverage (LEV) and liquidity (LIQ) are also added as measures of financial sustainability and short-term viability. H Index provides a measure of club sustainability based on the segregation of income variable. Additionally, two control variables are incorporated: the natural logarithm of total assets (LnTA) at the end of the fiscal year as a proxy variable (SIZEit) for club size; and variable (TIMEit), to capture the temporal effect of the learning process and adaptation to the new measures.

The impact of FFP regulations has translated into improvements in the financial balance of clubs 'to operate based on their own revenues' [33], and 'to encourage responsible spending' [33], and transparency has improved as a consequence of the implementation of TL 19/2013. Therefore, unlike Blavoukos et al. [78] and Dimitropoulos et al. [15], we argue that there are no substantial differences between the regulations 'intended' and actual outcomes.

The empirical analysis suggests that ROA has a positive impact on club transparency, while the more leveraged clubs with higher income concentrations are less transparent. Additionally, the transparency of football clubs is conditioned on their financial sustainability in the short (LIQ) and long term (SOLV). This fact is of special relevance if we consider that the main purpose of FFP regulations, and especially with the LFP's RCE, is 'to promote the solvency of clubs' [34]. Consequently, the general results confirm the hypothesis of the

availability of funds [56], according to which the financial performances of clubs influence social performance and, therefore, transparency. This fact is of special relevance since one of the LFP's RCE's specific aims is to 'improve the economic and financial capacity of clubs, increasing their transparency and credibility' [33,34].

The control' variable size (SIZEit), is positively related to transparency (INFUT). As such, our results confirm previous studies [15], which suggest that larger clubs are more sensitive to regulatory scrutiny [71,72], and they contradict the statement of TIE [43], that they are unrelated variables. The other control variable (TIMEit), manages to capture the gradual effect of the learning process and adaptation to the new measures.

This study adds a quantitative dimension to the traditional qualitative approaches to the analysis of institutional change in sports management [79]. In this regard, we argue that the processes observed have the potential to change the existing institutional logic that prioritizes sporting success over financial performance [2–4], thus contributing to a more transparent and rational approach to decision-making processes that directly concern the sustainability of clubs.

## 6. Conclusions

This study has examined the possible association between transparency and financial sustainability in Spanish professional football clubs. INFUT has been used as a measure of transparency and social performance, and UEFA's FFP and the LFP's RCE criteria as a measure of accountability and financial performance. The transparency and financial improvement suggest that clubs are sensitive to environmental regulations.

One of the main contributions of this study is the identification of a positive relationship between financial performance and transparency in Spanish football clubs. This result suggests that the implementation of TL 19/2013 and UEFA's FFP could generate a virtuous circle with the potential to change clubs' management logic in situations that can compromise their financial sustainability.

However, although both regulations were implemented simultaneously, some of their intended outcomes are taking longer to crystallize than others. It seems that making clubs transparent by providing information to stakeholders in an accurate, understandable, and accessible way is easier than making them accountable for their actions. It seems reasonable to think that the improvement of the club's financial situation and their sustainability is a long-term objective. Therefore, we shall need to continue observing future club behavior in terms of the transparency-financial sustainability, especially considering that FFP regulations are to be executed gradually, and that one of the limitations of this study is that no information exists before the implementation of TL 19/2013, as well as gaps created by TIE for the periods 2017 and 2018. Another limitation to the scope of this study is that only clubs that participate in European competitions are subscribed to the FFP regulations.

Despite recent progress in disclosure and financial practices in Spanish professional football, we believe that there is still a long way to go in terms of transparency and accountability. In order to examine new developments, future studies should also use qualitative methodologies to explain the isomorphism caused by environmental pressure. Another aspect to consider is the impact of the new regulations on the quality of accounting information on specific aspects, such as the transfer of players [80]. The relationship of transparency with non-financial performance measures could also be assessed [81]. Finally, the importance of other social variables which are complementary to transparency should be highlighted, such as the adoption of codes of ethics, good governance [82], environmental management, donations, and activities aimed at the social inclusion of minority or disadvantaged groups.

**Author Contributions:** R.U., J.C.G.-P., F.L.-M. and J.M.M. contributed equally to this work. All authors wrote, reviewed and commented on the manuscript. All authors have read and agreed to the published version of the manuscript.

**Funding:** This paper was made possible by funding from the Spanish Ministry of Science and Innovation, Project "CIRCULARTAX" Ref. PID2019-107822RB-I00.

**Institutional Review Board Statement:** Not applicable.

**Informed Consent Statement:** Not applicable.

**Data Availability Statement:** Data sharing not applicable.

**Acknowledgments:** We would like to appreciate the thoughtful and constructive advice provided by the reviewers, and especially the support of Néstor Le Clech from the Department of Economics and Business, National University of Quilmes, Argentina, and Emilio Martín Vallespín from the Accounting and Financial department, Zaragoza University.

**Conflicts of Interest:** The authors declare no conflict of interest.

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
