# Peer review of "Transparency and Accountability in Sports: Measuring the Social and Financial Performance of Spanish Professional Football"

_sustainability, doi:10.3390/su13158663_

Round 1
Reviewer 1 Report
Dear authors I found your paper an interesting an well written one. However I propose the following suggestions to furhther improve the manuscript: 1) In the conclusion I suggest to emphasize that the study is also limited with respect to the index employed to measure transparency as the use of other indixes could either confirm your findignds or lead to different outcomes. Therefore I would add that future studies may consider different measures of transparency 2) Cite the following papers: Lorenzo Neri, Antonella Russo, Marco Di Domizio & Giambattista Rossi (2021)Football players and asset manipulation: the management of football transfers in Italian Serie A, European Sport Management Quarterly, DOI: 10.1080/16184742.2021.1939397; Scafarto, V., & Dimitropoulos, P. (2018). Human capital and financial performance in professional football: The role of governance mechanisms. Corporate Governance: The International Journal of Business in Society, 18(2), 289–316. https://doi.org/10.1108/CG-05-2017-0096 You can use these references to suggest that future research directions which would include the governance mechanisms (e.g. as control variables) or other ways to measure transparency. Regards The reviewerAuthor Response
Dear reviewer,
Thank you very much for your receptivity to our article. We appreciate your thoughtful and constructive advice. In the new version of the paper, we have incorporated changes to address these suggestions and we believe that, as a result, the paper has been significantly improved. New text is in red.
Best regards

Reviewer 2 Report
This topic is very interesting and timely, as noted by the authors. A test of a new transparency index and whether it can be understood using existing financial and other variables.
- Go through another round of editing for grammar.
- Change FFPP to FFP.
- “of r” to “of” (p. 4)
- “y” to “and” (Table 2 note)
- Table 2: what is the difference between SP and PS or FP and PF?
- Each delta in equation (3) should have a subscript denoting that it’s different for each independent variable. Also, the HI is written as IH in the equation.
- In the paragraph in the middle of page 10, it says “…significant amount of equipment…” What is mean by equipment?
- Multicollinearity is likely an issue given the correlation findings among the independent variables. Key step is to run a variance-inflation factor just to see what it indicates, but more importantly to run the model without one of the highly correlated variables and see what the impact is on the other coefficients. Do signs or significance move around?
- Endogeneity may be an issue because ROA may be driven by some of the other control variables like solvency or leverage. Thus, one might really see a two-stage least squares situation where ROA = f(solvency, leverage, etc.), and then Transparency = g(fitted ROA) as the second stage regression. This can be done with FGLS also. Think hard about the possibility that leverage drives ROA, as the point of leverage is to drive ROE (Equity), which is part of Assets.
Author Response
Dear reviewer,
Thank you very much for your comments. We appreciate your thoughtful and constructive advice. Below, we try to respond to each of the issues raised in your review. In the new version of the paper, we have incorporated changes to address some of these suggestions and we believe that, as a result, the paper has been significantly improved. New text is in red and the text has been deleted is marked with a crossed-out line.
Best regards

Reviewer 3 Report
I thank the authors for this proposal. The subject is original, very interesting and the proposal is of very good level. However, I invite them to rework their proposal according to the remarks in the attached file.

Author Response

(The authors gave the same response as above.)

Round 2
Reviewer 2 Report
The author(s) satisfactorily addressed my concerns.
Reviewer 3 Report
Thank you for your comments
Congratulations for this beautiful article
Sincerely